# Sharing Studies between 5G IoT Networks and Fixed Service in the 6425–7125 MHz Band with Monte Carlo Simulation Analysis

**DOI:** 10.3390/s22041587

**Published:** 2022-02-18

**Authors:** Alexander Pastukh, Valery Tikhvinskiy, Evgeny Devyatkin, Aigul Kulakayeva

**Affiliations:** 1Radio Research and Development Institute, 105064 Moscow, Russia; vtniir@mail.ru (V.T.); deugene@list.ru (E.D.); 2International Information Technology University, Almaty 050000, Kazakhstan; aigul_k.pochta@mail.ru

**Keywords:** IMT-2020, 5G IoT, spectrum sharing, spectrum management, spectrum engineering, compatibility studies, Monte Carlo simulation, SEAMCAT, beamforming, frequency regulation, ITU, WRC, coordination, signal-to-interference, interference-to-noise

## Abstract

This work presents sharing studies between 5G networks and point-to-point fixed service in the 6425–7125 MHz band. In this research, we provide simulations of interference from 5G downlink and uplink to fixed service in the frequency band 6425–7125 MHz. We evaluated several scenarios of interference, which include cross-border scenarios, as well as scenarios of interference within the borders of one administration. The obtained results of this work are presented as protection distance and frequency offsets that are required in order to achieve compatibility between 5G and FS in the 6425–7125 MHz band. The spectrum engineering techniques presented in this research can help different companies and regulatory administrations in their spectrum management and frequency regulation activities and seriously improve the efficiency of implementation for 5G technologies.

## 1. Introduction

This article is a significant extension of the previously done studies [1], related to sharing of the frequency band 6425–7125 MHz, between 5G and fixed service (FS). While previous studies focused on cross-border scenarios only, this study includes new compatibility research, related to sharing the frequency band 6425–7125 MHz, between 5G and FS at a national level, which includes interference scenarios where the receiving fixed service station is located within the borders of the 5G network cluster. Additionally, adjacent channel interference simulations were done and protection frequency offsets were proposed. This article also analyzes different approaches that national administrations can use in order to achieve compatibility between 5G and FS in the 6425–7125 MHz frequency band. Lastly, in this article we provided more detailed methodology of how the interference from 5G user equipment was simulated.

The modern development of IoT technologies is closely related with the emergence of fifth generation (5G) technologies. Further, 5G provides a set of specifications that includes ultra-reliable low-latency communications (URLLC) and massive machine type communications (mMTC). Together, URLLC and mMTC allow the significant enhancement of IoT technologies and bringing them to the next stage of development, effectively complementing the IoT ecosystem [2,3]; 5G is a pillar of digital transformation. According to GSMA (article «Capacity to Power Innovation: 5G in the 6 GHz band), its integration into our lives and work has the potential to impact communities and economies, and as it delivers transformational services, it can boost global GDP by USD 2.2 trillion. In 2019, mobile technologies and services generated 4.7% of GDP across the globe. By 2024, the contribution is predicted to increase to 4.9% of GDP.

According to analytical forecasts, in 2024, the revenues of the operators in the IoT segment will reach USD 8 billion (compared to USD 525 million in 2020). It is expected that by 2023, 5G IoT will reach 3.5 billion connections, and by 2025, 5G-IoT will share 70% of all IoT connections worldwide.

To achieve such ambitious goals, it is crucial to have sufficient spectrum resources to provide enough coverage and throughput of the IoT networks. Today, regulatory bodies worldwide are investigating the use of the 6425–7125 MHz band for 5G technologies. This spectrum range is very attractive, since it’s a mid-band frequency range that combines strong sides of FR1 and FR2 bands, effectively balancing between coverage and capacity, providing the perfect environment for 5G connectivity.

Extending the bandwidth of 5G through harmonization of 6425–7125 MHz will open new opportunities for the IoT networks and significantly reduce the cost of IoT network deployments, making it more affordable.

The World Radiocommunication Conference 2023 (WRC-23) will play an important role in determining future access to the 6425–7125 MHz band. Today the ITU Radiocommunication Sector (ITU-R) considers the identification of the 6425–7025 MHz (Region 1) and 7025–7125 MHz (globally) for IMT, including possible allocations to the mobile service on a primary basis in these bands. Therefore, ITU members worldwide are invited to do sharing and compatibility studies to protect services with an existing primary allocation, without imposing additional regulatory or technical constraints.

At present there are no commercial solutions of 5G base stations or user equipment and characteristics of them are preliminary provided by 3GPP and ITU-R (see 3GPP TS.38.104 v. 16.6.0 (2020-12) «NR; Base Station (BS) radio transmission and reception», 3GPP TS.38.101-1 v.16.6.0 (2020-12) « NR; User Equipment (UE) radio transmission and reception; Part 1: Range 1 Standalone» and Document 5D/TEMP/422). The architecture options of 5G base stations that are defined in specifications 3GPP TS 38.104 and 3GPP TS 38.401, as well as in ITU-T Rec. G.8300 (05/2020), allow using the integration of fronthaul and backhaul in one base station for the bandwidth 6425–7125 MHz (FR1). The architecture solutions that integrate both fronthaul and backhaul in one BS do not influence the issue of spectrum sharing and electromagnetic compatibility, since the architecture is related to network design rather than compatibility issues. Therefore, the obtained results by the authors are applicable for any architecture solution of 5G base stations.

Currently, sharing studies with the following services are conducted for 6425–7125 MHz band in ITU-R: fixed service, fixed-satellite service (Earth-to-space), fixed-satellite service (space-to-Earth), space operation service, and space research service. The allocation of the services is presented in Figure 1 below:

The fixed service point-to-point systems provide radio communication by a link, for instance, radio-relay link, between two stations located at specified points. Fixed service point-to-point stations are utilized worldwide, in almost every country and, thus, should be taken into account when deploying 5G networks.

As can be seen above, unlike other services, fixed service is allocated across the entire band, as well as to the adjacent bands; therefore, compatibility between 5G and fixed service is critically important. It should be noted that when it comes to the spectrum sharing between countries, sharing between 5G and fixed service is a subject of coordination, since for many applications, the location of the fixed transmitters and receivers are known by the regulatory bodies and transmission characteristics are regulated. When it comes to the sharing between different services within one administration, compatibility issues are a subject of national regulation bodies that use different techniques to achieve compatibility within one city or rural zone [3,4].

## 2. State of Art

The study which is the basis of the present research in the band 6425–7125 MHz [1] does not contain an interference scenario with the shared location of 5G networks and FS, additionally, that the study doesn’t contain adjacent channel interference simulation and doesn’t refer to the national level regulation focusing solely on the cross-border scenario. Thus, this research complements the previous study as well as takes into account conditions that were not considered in the related studies for the millimeter-waves band.

When it comes to simulation and spectrum management techniques and approaches, this work takes into account various works on that topic [4,5] that are commonly used in the studies in such organizations like ITU-R, CEPT and many others.. Earlier works on the matter of sharing studies have introduced the need defining compatibility conditions between 5G networks or networks of legacy standards and fixed service stations [6,7,8,9,10,11,12,13,14], at the same time these studies cannot be fully applied to 6425–7125 MHz because of the entirely different deployment between those two bands since propagation losses between 6 GHz and millimeter waves are significantly different, also millimeter-wave base stations have shorter antenna heights and the different number of antenna configurations which affect dramatically on the interference levels. In addition, these works include only interference study from BS but do not include the study of interference from UE, usually, commonly it is explained that since 5G uses TDD there is no need to study UE interference since it’s normally negligible compared to the interference from BS, the authors of this paper believe that interference from UE needs to be studied anyway for the reason that in many cases operation of BS may be restricted depending on the regulators’ needs, whereas UEs normally cannot be regulated, and in some local scenarios the interference from UEs can be comparable with BS interference. 

## 3. Materials and Methods

The studies considered co-channel and adjacent channel interference cases. The simulation takes into account interference from both uplink and downlink of 5G. Figure 2 below presents a typical scenario of 5G downlink and uplink interference to a point-to-point fixed service station [6,7,8,9,10,11,12,13,14].

The fixed service parameters were configured according to and summarized in Table 1. The protection criterion for the fixed service used was *I*/*N* = −10 dB for 20% of the time according to the Recommendation ITU-R F.758. For sensitivity analysis, additional protection criterion *C*/*I* = 30 dB was considered which corresponds to the 64QAM modulation order that is the most typical for fixed service point-to-point stations in the 6425–7125 MHz frequency band. This criterion was derived from the ITU-R 5C/39 document.

The link length of the fixed service stations was 10 km for the stations with 20 m antenna heights and 38 km for the stations with 60 m antenna heights. In simulations different deployment of FS receivers was considered, the victim FS receiver was placed with both main lobe and side lobe pointed to the 5G network.

In order to simulate 5G network, the parameters summarized in Table 2 and Table 3 were configured.

It should be noted that ohmic losses are already included in the antenna gain value with the antenna configuration that is used, while conducted power assumes 16 × 8 × 2 elements (i.e., power per H/V polarized element).

To calculate interference taking into account off-channel rejection in accordance with Rec. ITU-R SM.337, the following spectrum emission masks (Figure 3) of 5G base stations and user equipment were configured. For the adjacent channel calculation, 5G center frequency was 70 MHz offset from the center frequency of the FS station.

The 5G network was simulated as a cluster that contained 19 BS, each BS covered 3 sectors and each sector contained 3 simultaneously active users [15]. For the 5G BS antennas beamforming that was adopted, using the antenna pattern according to Rec. ITU-R M.2101, this is presented in Figure 4.

The BS beamforming precoding is used and multiple spatially directive signals are transmitted simultaneously [16]. In simulations, the emission consisted of three directive beams pointing to every user in each sector; the output power of the beam was evenly split to each user as shown in Figure 5 below:

The value 41 dBm/33 MHz was obtained using the following expression:*P*_33 MHz_ = 22 + 10 ∗ log(8 × 16 × 2) − 10 log(3)(1)
where 22 (dBm) is conducted power per element, 8 × 16 × 2 (equals 256 in total) is the number of antenna elements (i.e., power per H/V polarized element) and 3 is the number of beams.

The simulations adopted a TDD duplex with 75% BS and 25% UE activity factor. It should be noted that TDD was synchronized and all 19 BS were active/inactive simultaneously. Such synchronization is the most realistic case since asynchronous TDD with Bernoulli distribution would lead to inter-cell interference within the 5G network. The network loading factor used in this study was configured as 50% which normally represents a small area analysis for the worst-case scenario. However, it should be noted that in a small area with a few IMT transmitters, if the loading approaches 50%, then the IMT network performance would not be sufficient (e.g., dropped calls will occur) and more capacity will need to be installed. Therefore, in practice, the network loading factor will most likely be less than 50%. This network loading factor was chosen in accordance with ITU-R Working Party 5D characteristics for IMT-2020 that are used in the current study cycle.

The propagation was calculated according to Rec. ITU-R P.2001 and 20% of the time was configured. The simulation scenarios take into account additional clutter losses which were configured according to Rec. ITU-R P.2108.

For the 5G network urban deployment scenario 65% of locations were configured, whereas for the suburban deployment scenario 15% of locations were configured.

For the fixed service, no clutter was considered for the 60 m height stations, whereas for the 20 m height 50% of locations were considered; it should be noted that for the 20 m stations 50% of locations were considered only for the interfering link, whereas for the wanted link no clutter has been considered since point-to-point stations are designed so that that Fresnel zone is free of any obstructions.

In the studies, a Monte Carlo simulation with 2000 events is performed to assess the linear average interference-to-noise ratio (*I*/*N*) and signal-to-interference ratio (*C*/*I*) when analyzing interference from the cluster of 5G cells to the FS receiver.

After each simulation, the distance between the FS receiver and the edge of the 5G cluster was changed and the simulation was re-run until the above-mentioned criteria were fulfilled. Such an approach allowed calculating required separation distances between the 5G network and FS receivers.

The interference was evaluated through the SEAMCAT simulator [17,18]. Figure 6 below shows the example of interference simulation from 5G to FS receiver in SEAMCAT where the interference from BS 5G and UE 5G is simulated.

The interference from the *i*th active 5G BS or the *j*th active UE to the FS receiver can be calculated by the following equation [19]:*I_BS/UE_* = *P_TX_* + *G_IMT_* + *G_FS_* − *L_p_* − *L_xpr_*
(2)
where, *P_TX_* is the transmitted power of the IMT BS or UE (dBm); *G_IMT_* is the transmit antenna gain of the IMT BS or UE towards the victim receiver (dBi); *G_FS_* is the receive antenna gain of the FS towards the interfering station (dBi), *L_p_* is the propagation loss from the IMT BS/UE to the FS receiver (dB), *L_xpr_* is the polarization loss (dB).

At each simulation event interference from every beam of 5G was calculated. Aggregate interference was calculated using the expression below [19]:(3)$$\frac{I}{N}\left[\mathrm{dB}\right]=10\log_{10}\left({\mathrm{Pr}}_{\mathrm{BS}\_\mathrm{TDD}}\sum_{i}10^{\frac{I_{BS}\left(i\right)}{10}}+{\mathrm{Pr}}_{\mathrm{UE}\_\mathrm{TDD}}\sum_{j}10^{\frac{I_{UE}\left(j\right)}{10}}\right)-\left(D+NF+10\log\left(B\right)\right)$$
where, *I_BS_* (*i*) and *I_UE_* (*j*) are the interference from *i*th active IMT BS or *j*th active UE to the FS receiver, respectively (dBm); *P_rBS_TDD_* is the TDD activity factor of the IMT BS; *P_rUE_TDD_* is the TDD activity factor of the UE; *D* is receiver noise power density (dBm/Hz); *NF* is receiver noise figure (dB), *B* is receiver channel bandwidth (Hz).

The Figure 7 below shows the example of interference calculation from one of the beams of 5G BS at one simulation event and calculating aggregate interference.

Wanted signal of the FS station receiver can be calculated by the following Equation [19]:*C_FS_* = *P_FSTx_* + *G_FSTx_* + *G_FSrx_* − *L_p_* − *L_xpr_* − *A_fading_*(4)
where, *P_FST_*_x_ is the transmitted power of the FS transmitter (dBm), *G_FSTx_* is the transmit antenna gain of the FS transmitter towards FS receiver (dBi), *G_FSrx_* is the receive antenna gain of the FS receiver towards the FS transmitter (dBi), *L_p_* is the propagation loss from the FS transmitter to FS receiver (dB) *L_xpr_* is the polarization loss (dB) and *A_fading_* is loss due to fading effects.

Total *C*/*I* taking into account aggregate interference from 5G BS and UE can be calculated using the following equation [19]:(5)$$\frac{C}{I}\left[\mathrm{dB}\right]=C_{FS}-10\log_{10}\left({\mathrm{Pr}}_{\mathrm{BS}\_\mathrm{TDD}}\sum_{i}10^{\frac{I_{BS}\left(i\right)}{10}}+{\mathrm{Pr}}_{\mathrm{UE}\_\mathrm{TDD}}\sum_{j}10^{\frac{I_{UE}\left(j\right)}{10}}\right)$$
where, *C_FS_* is wanted signal of the FS station (dBm); *I*_BS_ (*i*), *I*_UE_ (*j*) is the interference from *i*th active IMT BS or *j*th active UE to the FS receiver, respectively (dBm); *Pr_BS_TDD_* is the TDD activity factor of the IMT BS, *Pr_UE_TDD_* is the TDD activity factor of the UE.

When calculating interference from 5G Uplink, power control for the UE was implemented. Power control is a significant technical feature of IMT systems. The uplink cell capacity in OFDMA-based systems is constrained by interference levels from other UEs. UE power output levels are adjusted to maintain minimum interference and to ensure cell edge coverage. Power control can be combined with frequency–domain resource allocation strategies to enhance cell edge performance as well as improve spectral efficiency. The following algorithm was used to implement power control [20,21]:*P_PUSCH_* (*i*) = min(*P_CMAX_*, 10 log_10_ (*M_PUSCH_* (*i*)) + *P_O_PUSCH_* (*j*) + *α*(*j*) · *PL*)(6)
where, *P_PUSCH_* is transmit power of the terminal (dBm), *P_CMAX_* is maximum transmit power (dBm), *M_PUSCH_* is number of allocated resource blocks (RBs), *P_0_PUSCH_* is power per RB used target value (dBm), α is balancing factor for UEs with bad channel and UEs with good channel, PL is path loss for the UE from its serving BS (dB).

## 4. Results

Based on the studies of co-channel interference, protection distances between the edge of the 5G network and FS receiver for two criteria *(I*/*N* and *C*/*I*) were derived. The protection distances are summarized in the tables and charts below.

Table 4 below shows the separation distances between the FS station and the 5G cluster required to protect FS, according to *I*/*N* = −10 dB protection criteria for main lobe and side lobe interference scenarios.

Figure 8 below contains a chart with protection distances required between 5G and 60 m height FS receivers, based on *I*/*N* = −10 dB protection criteria.

Table 5 below shows the separation distances between the FS station and the 5G cluster required to protect FS, according to *C*/*I* = 30 dB protection criteria for main lobe and side lobe interference scenarios.

The Figure 9 below contains a chart with the protection distances required between 5G and 60 m height FS receivers, based on *C*/*I* = 30 dB protection criteria.

Taking into account that above-mentioned results for co-channel interference, it can be seen that we cannot place an FS receiver within the same areas as 5G networks, for the same deployment scenario where 5G and FS are located inside the same area frequency. Separation was used and the simulation was done based on *I*/*N* and *C*/*I* protection criteria. In order to calculate adjacent channel interference, adjacent channel selectivity (ACS) of FS stations and adjacent channel leakage ration (ACLR) of 5G need to be used. ACS of the FS station is provided in Table 1, whereas the ACLR of 5G can be obtained from spectrum emission masks, provided in Figure 3. Here, the ratio of the total interference between adjacent channels is given by the adjacent channel interference ratio (ACIR), hence, the following:(7)ACIR=10log(1110^(ACR/10)+110^(ACLR/10))
where ACS is adjacent channel selectivity (dB) and ACLR adjacent channel leakage ratio (dB). Note that while ACR is provided as a single value, ACLR is mostly provided in transmitting mask and usually varies, depending on the frequency offset. In case ACLR varies in spectrum emission mask, to take into account the offset, the following expression should be used:(8)$$FDR(\Delta f)=10\log\frac{\overset{\infty }{\underset{0}{\int }}P(f)df}{\overset{\infty }{\underset{0}{\int }}{P(f)|H(f+\Delta f)|}^{2}df}$$
where *P*(*f*) is the power spectral density of the interfering signal equivalent intermediate frequency (W/Hz), *H*(*f*) is the frequency response of the receiver, depending on Δ*f* = *ft* − *fr* (MHz), where *ft* interferer tuned frequency *fr* receiver tuned frequency.

Thus, adjacent channel interference can be calculated using the following expression:*I_BS_*_/*UE*_ = *P_TX_* + *G_IMT_* + *G_FS_* − *L_p_* − *L_xpr_* − *FDR*(Δ*f*)(9)

The following *I*/*N* curves in Figure 10 were obtained for the deployment scenario when 5G network and FS receiver were located within the same area, with adjacent channel frequency separation, where the curves mean the following:The red curve is for urban deployment, using 60 m FS stations;The blue curve is for urban deployment, using 20 m FS stations;The green curve is for suburban deployment, using 60 m FS stations;The yellow curve is for suburban deployment, using 20 m FS stations;

As can be seen from the figure, *I*/*N* criteria was exceeded, from 10 to 45 dB, for every deployment scenario of 5G and for every height of the FS station. The results clearly indicate that co-location of 5G network and FS stations, even in adjacent channels, might be problematic if compatibility is estimated using *I*/*N* criterion.

The following *C*/*I* curves in Figure 11 were obtained for the deployment scenario, when 5G network and FS receiver were located within the same area, with adjacent channel frequency separation. Where the red curve is for urban deployment using 60 m FS stations, the blue curve is for urban deployment using 20 m FS stations, the green curve is for suburban deployment using 60 m FS stations, and the yellow curve is for suburban deployment using 20 m FS stations.

As can be seen from the above figure, *C*/*I* criteria is met in 100% of snapshots, and margins are from 5 to 50 dB. The results indicate that planning 5G and FS co-location scenarios, using *C*/*I* criteria, allows for compatibility in shared locations.

## 5. Discussion

The obtained results indicate that compatibility between 5G and FS depends on the protection criteria that have been chosen. Traditionally, *I*/*N* criteria is used in many studies, and may be considered as a conservative approach. At the same, the *I*/*N* criteria is pretty stringent and does not always reflect the real-case scenario. Given that fixed service point-to-point stations have fixed locations, and their antenna patterns are pointed towards each other’s main lobes, it is possible to calculate the wanted signal with high precision. Therefore, using *C*/*I* criteria allows us to consider a more realistic case. *C*/*I* criteria largely depends on the modulation order and coding rate used [22]. In this study, *C*/*I* criteria was used for the highest order modulation (64 QAM) that is supported by FS, in the 6425–7125 MHz frequency band. This means that for the lower modulation orders, *C*/*I* criteria would be lower than 30 dB and, thus, for lower modulation orders, the FS system would have an even larger *C*/*I* margin.

It should be kept in mind that for cross-border scenarios, it is very rare when an FS station is oriented with its main lobe towards the border, a side-lobe interference scenario is much more common. At the same time, main lobe scenarios may occur as well and taking into account that the typical link length of FS stations is 38 km for the 60 m height antennas, and 10 km for the 20 m height antennas, it can be concluded that protection distances based on *C*/*I* are enough to provide compatibility in cross-border scenarios. Figure 12 below illustrates why protection distances lower than link length provide compatibility between FS and 5G.

We would also like to note, that using territorial or frequency separations are not the only ways to achieve compatibility between 5G and FS in cross-border scenarios. Many other techniques can be applied, for example, the administrations may agree on sectorial restrictions of 5G BS; in particular, azimuths and elevation angles of 5G antennas may be restricted in the direction of the neighboring administration.

At the national level, 5G operators may agree with an FS operator to change its FS station to the station that operates in another frequency range; such substitution can be compensated by the involved 5G operator and is a very common way to solve compatibility issues at the national level. Therefore, regulators are not limited to the approaches proposed in this paper, although other solutions require the analysis of a concrete case.

## 6. Conclusions

The in-band sharing studies between the FS and 5G showed that when using *I*/*N* protection criterion, the separation distance of the FS receivers from the edge of the 5G network for the main lobe interference scenario should be from 10 to 62.5 km, depending on the FS station antenna height and 5G deployment scenarios. For the side lobe scenario, the protection distance should be less than 1 to 10 km, depending on the FS antenna height and 5G deployment scenario. When using the *C*/*I* protection criterion, distances should be from 3 to 32 km for the main lobe scenario and less than 1 km for the side lobe scenario. The protection distances based on *C*/*I* are enough to provide compatibility in a worst-case scenario, since the protection distances are shorter than the FS link lengths.

When 5G and FS are located inside the territory of one administration, frequency separation can be used, and based on the results of *I*/*N* criteria, it can be seen that *I*/*N* values exceed the protection criteria *I*/*N* = −10 dB. The results for additional protection criterion *C*/*I* = 30 dB indicate *C*/*I* ratio is fulfilled with sufficient margins for every modulation order; thus, when using *C*/*I* protection, criteria compatibility can be achieved using frequency separation between the 5G and FS channels in co-location scenarios.

## Figures and Tables

**Figure 1 sensors-22-01587-f001:**
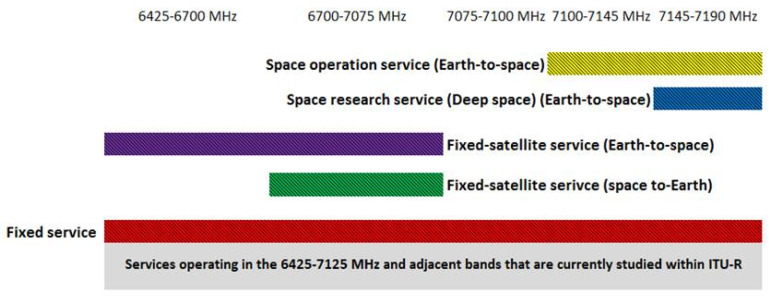
Services operating in the 6425–7125 MHz and adjacent bands.

**Figure 2 sensors-22-01587-f002:**
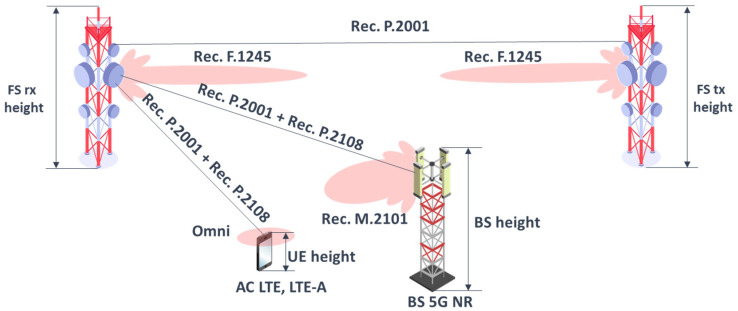
Typical interference scenario between 5G and FS station.

**Figure 3 sensors-22-01587-f003:**
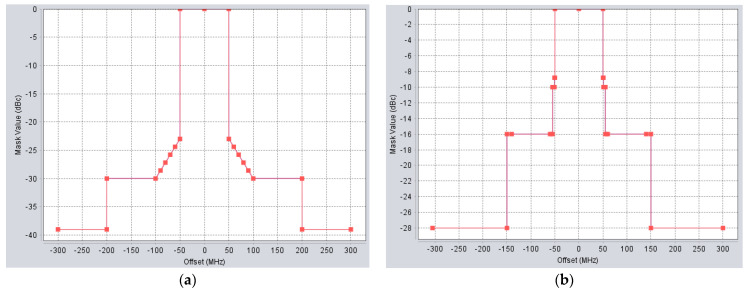
Spectrum emission masks of 5G transmitters (**a**) BS spectral emission mask; (**b**) UE spectral emission mask.

**Figure 4 sensors-22-01587-f004:**
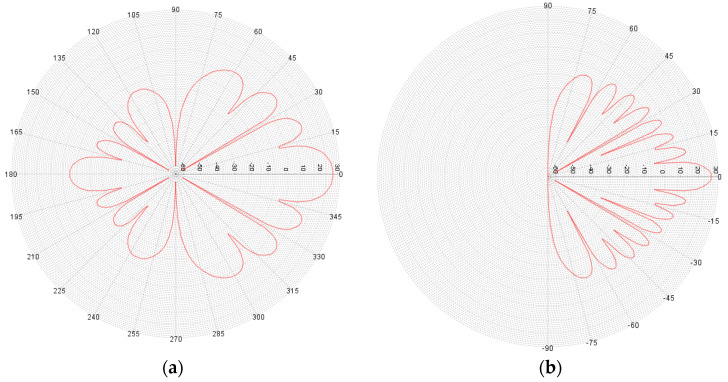
Antenna gain pattern (**a**) Horizontal plane; (**b**) Vertical plane.

**Figure 5 sensors-22-01587-f005:**
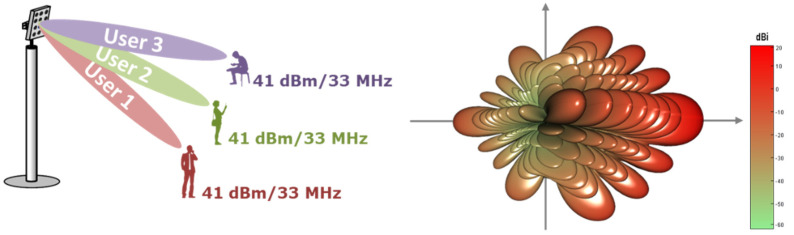
Beam distribution used in simulations and gain pattern of each beam.

**Figure 6 sensors-22-01587-f006:**
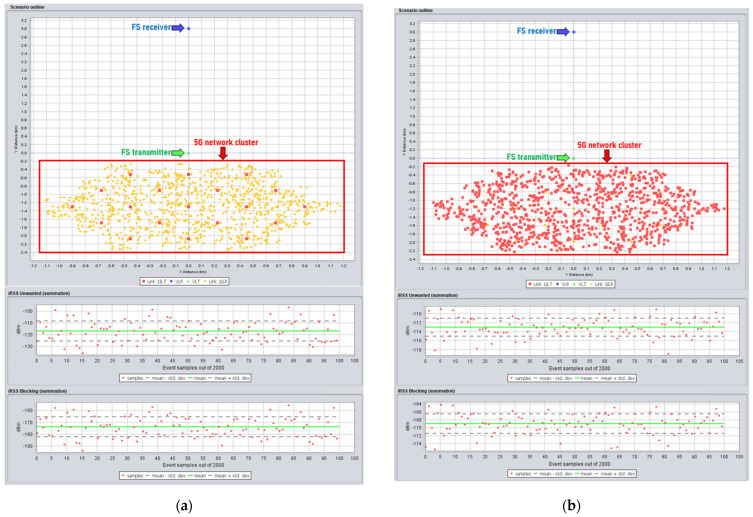
Example of SEAMCAT simulation between IMT-2020 and FS station (**a**) interference modeling from BS 5G; (**b**) interference modeling from UE 5G.

**Figure 7 sensors-22-01587-f007:**
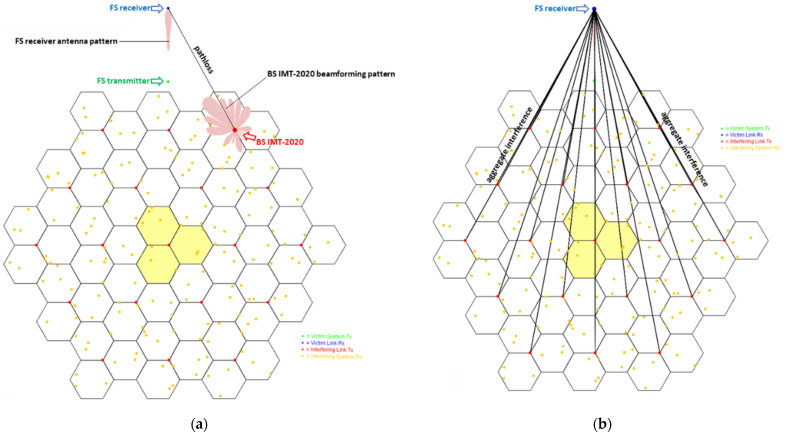
Example of interference calculation at one snapshot (**a**) Single-entry interference calculation; (**b**) Aggregate interference calculation.

**Figure 8 sensors-22-01587-f008:**
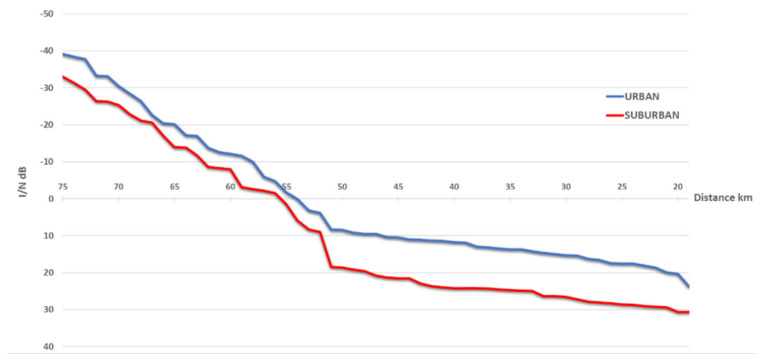
Protection distance between 5G and 60 m height FS station based on *I*/*N* = −10 dB criteria (main lobe).

**Figure 9 sensors-22-01587-f009:**
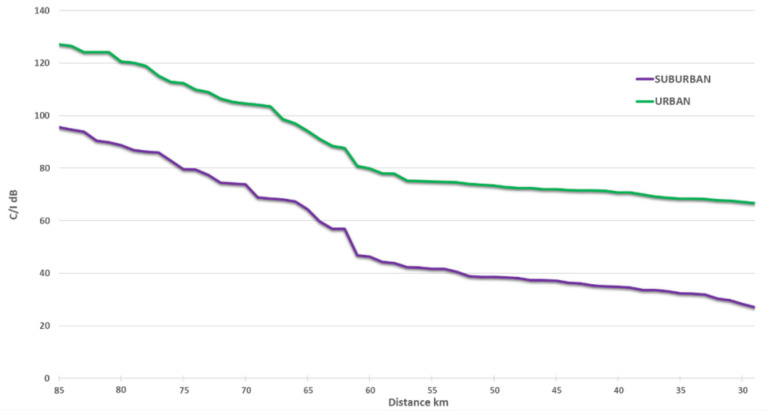
Protection distance between 5G and 60 m height FS station based on *C*/*I* = 30 dB criteria (main lobe).

**Figure 10 sensors-22-01587-f010:**
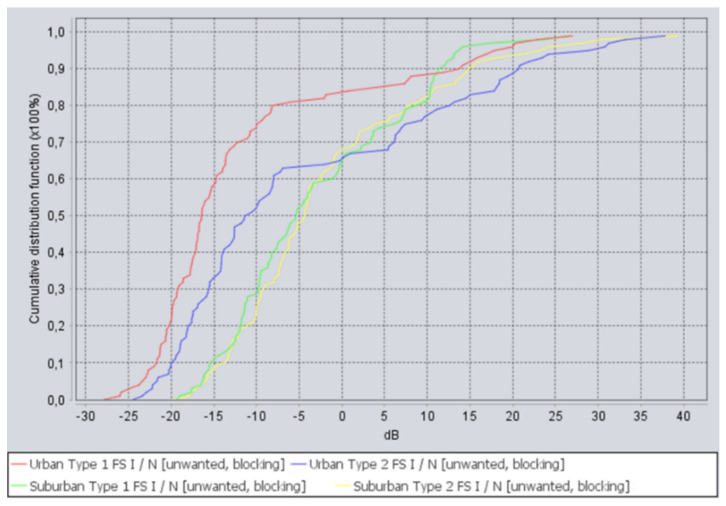
*I*/*N* curves for adjacent channel interference when 5G network and FS receivers are located in the same area.

**Figure 11 sensors-22-01587-f011:**
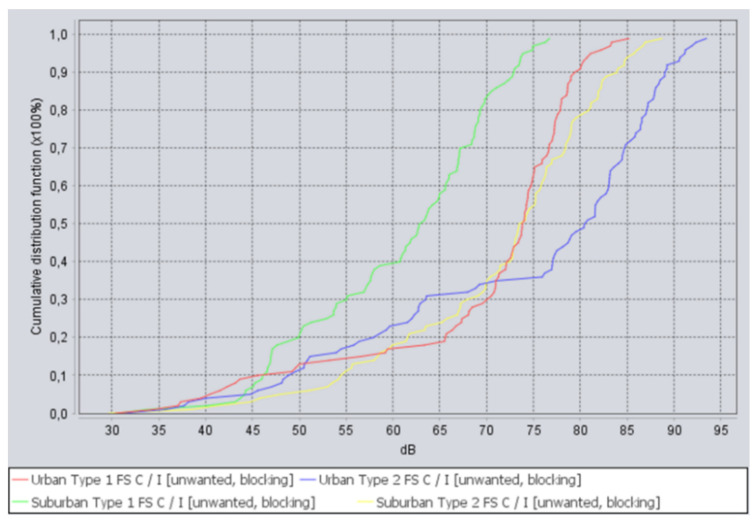
*C*/*I* curves for adjacent channel interference when 5G network and FS receivers are located in the same area.

**Figure 12 sensors-22-01587-f012:**
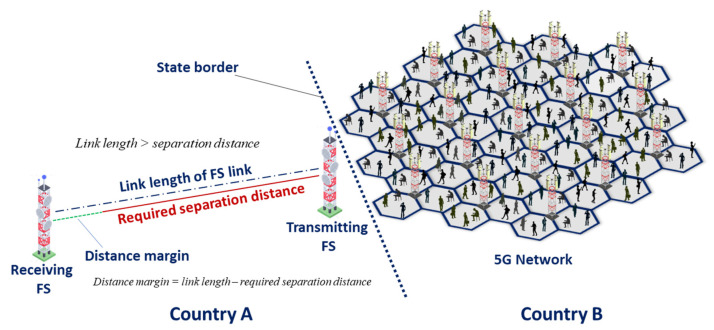
Cross-border scenario case.

**Table 1 sensors-22-01587-t001:** FS characteristics.

Parameter	Type 1	Type 2
Frequency	6425–7125 MHz	6425–7125 MHz
Link length	38 km	10 km
Propagation model	ITU-R P.2001-4	ITU-R P.2001-4
Antenna heights	60 m	20 m
Antenna gain	38–39.5 dBi	36 dBi
Noise figure	5 dB	5 dB
Antenna pattern	ITU-R F.1245	ITU-R F.1245
Channel bandwidth	40 MHz	40 MHz
Adjacent channel selectivity	25 dB	25 dB

**Table 2 sensors-22-01587-t002:** 5G base station characteristics.

Parameter	BS Urban	BS Suburban
Frequency	6425–7125 MHz	6425–7125 MHz
Channel bandwidth	100 MHz	100 MHz
Cell radius	300 m	600 m
Antenna mechanical downtilt	10°	6°
Antenna height	18 m	20 m
Antenna configuration	8 × 16 elements	8 × 16 elements
Antenna pattern	ITU-R M.2101	ITU-R M.2101
Antenna gain	5.5 dBi	6.4 dBi
Conducted power per element	22 dBm	22 dBm
Ohmic losses	3 dB	3 dB

**Table 3 sensors-22-01587-t003:** 5G user equipment characteristics.

Parameter	UE
Frequency	6425–7125 MHz
Channel bandwidth	100 MHz
Antenna height	1.5 m
Antenna pattern	Omnidirectional
Antenna gain	−4 dBi
Conducted power	23 dBm
Body loss	4 dB

**Table 4 sensors-22-01587-t004:** Separation distances between the edge of 5G network and FS based on I/N = −10 dB protection criteria for co-channel interference.

5G Deployment Scenario	Type 1	Type 2
Suburban (main lobe)	62.5 km	28 km
Urban (main lobe)	58.5 km	10 km
Suburban (side lobe)	10 km	<1 km
Urban (sidelobe)	3 km	<1 km

**Table 5 sensors-22-01587-t005:** Separation distances between the edge of 5G network and FS receiver based on *C*/*I* = 30 dB protection criteria for co-channel interference.

5G Deployment Scenario	Type 1	Type 2
Suburban (main lobe)	32 km	8 km
Urban (main lobe)	5 km	3 km
Suburban (side lobe)	<1 km	<1 km
Urban (side lobe)	<1 km	<1 km

## Data Availability

The data presented in this study are available upon request from the corresponding authors.

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
