# Peer review of "Sharing Studies between 5G IoT Networks and Fixed Service in the 6425–7125 MHz Band with Monte Carlo Simulation Analysis"

_sensors, 2022, doi:10.3390/s22041587_

Round 1
Reviewer 1 Report
General evaluation
The paper presents the results of compatibility study between 5G networks and point-to-point fixed service in the 6425-7125 MHz frequency band. The attention was paid to the interferences coming from 5G network. The simulation experiments were correctly designed and included co-channel and adjacent channel interferences. The protection distances between 5G network and point-to-point fixed service were derived and discussed.
The paper is well written. There are some minor errors (some typical examples are given below) Please, correct all such errors.
- Regarding the scientific content:
- It seems that the Figure 8 doesn’t correspond to the results given in the Table 4. For the suburban area the red curve crosses the protection criteria I/N=-10 dB for the distance of 62,5 km, while for the urban area the blue curve crosses this criteria for the distance of 58,5 km. The results of simulations given in the Table 4 (for main lobe) are as follows: 64 km for the suburban area and 56 km for the urban area. Please correct the Table 4 or use the proper chart.
- The same with the Figure 9 and the Table 5. Does the Figure 9 present results for the main lobe? Curves in the Figure do not cross the protection criteria (C/I=30 dB) at the distances given in the Table 5. Please explain how to interpret the chart or use the proper chart or correct the Table 5.
- Please, provide more detailed discussion of results presented in Figure 10 and Figure 11,
- Please, add explanation for co-channel interference simulations. Was the center frequency for the 5G signal (100 MHz bandwidth) exactly the same as for FS signal (40 MHz bandwidth)?
- Regarding the formatting and presentation:
- Please, numerate all equations.
- Please, correct descriptions of deployment scenarios in the Table 4. The scenarios are described as “main lobe” four times, while two scenarios should be “side lobe”.
- The same for the Table 5 - two scenarios should be “side lobe”.
- Please, correct the equation given in line 187 (wanted signal of the FS station). It can’t be IBS/UE, because IBS/UE represents the noise level from 5G BS or UE. Wanted signal of the FS station should be represented by CFS as it is used in the equation in line 195.
- The details given in the Figure 6 and the Figure 7 are not well visible. Please, use better quality pictures.
- Please, add the references:
- in line 97 – for selection of the protection criteria I/N = -10 dB,
- in line 98 – for selection of the protection criteria C/I = 30 dB,
- in lines 46-48 – for the foreseen increase of GDP,
- in line 141 – for the network loading factor.
- In the Figure 5:
- give the value of the bandwidth for the User 3,
- make sure, that the value (41dBm) is correct and explain how it was calculated.
- In line 142 correct the Recommendation number – it should be ITU-R P.2001.
- In the legend of the Figure 10 and the Figure 11 - what is the meaning of numbers (2, 22, 222) given at the end of square brackets? Are these numbers necessary?
- Grammar and style:
Line 46 – “US$2.2 trillion1” must be corrected as “US$2.2 trillion”,
Line 91 – “both interference from both uplink and downlink of 5G” must be corrected as “interference from both uplink and downlink of 5G”,
Line 118 – “was simulated according to as a cluster” must be corrected as “was simulated as a cluster”
Line 139 – (e.g. dropped calls will occur, etc.) must be corrected as (e.g., dropped calls will occur)
Line 181 – “the example of calculation interference from” must be corrected as “the example of interference calculation from”
Line 186 – “equation[19]” must be corrected as “equation [19]”
Line 191 – “the polarization loss (dB) Afading is” must be corrected as “the polarization loss (dB) and Afading is”
Line 202 – “UE power output levels are adjusted to maintain minimum interference and are adjusted to ensure cell edge coverage” must be corrected as “UE power output levels are adjusted to maintain minimum interference and to ensure cell edge coverage”.
Line 216 – “for two criteria (I/N and C/I).” must be corrected as “for two criteria (I/N and C/I) were derived.”
Line 226 – add “main lobe” at the end of the Figure 8 caption (if it presents results for main lobe)
Line 230 – “the edge of 5G network 5G and FS receiver” must be corrected as “the edge of 5G network and FS receiver”
Line 236 – the same as in line 226
Lines 237-240 – the sentence is too long and should be divided.
Lines 264-271 - the sentence is very long and not clear. This should be corrected.
Lines 300-305 - the sentence is very long and not clear. This should be corrected.

Author Response
Thank you for your review, I have edited the article according to your review.
The grammar typos and equations are corrected.
I added the formula on how to calculate -41 dBm/33 MHz, please note that additional information under table 2 was added which would also clarify how to calculate it.
The references for protection criteria I/N and C/I are added in the text.
Regarding figures 6 and 7 I tried to enlarge them I hope they are more clear right now, the reason why they were smaller is that there were some limitations on the article size.
Regarding inconsistency of I/N and C/N in Figures 8 and 9, I corrected Table 4.
Added formulas that can be used to calculate co-channel interference as well as added the information of the center frequency.
The legend in figures 10 and 11 is corrected.
The description of figures 10 and 11 is provided after each figure.
I corrected the texts in discussions and conclusions, additionally, I also added a picture to the discussions to make everything clearer.
Please see the following new version of the article! If something hasn't been corrected, please don't hesitate and tell me, thank you in advance.
Best regards,
Alexander

Reviewer 2 Report
The paper presents an analysis for haring fronthaul i backhaul in 5G in the band of 6.4-7.1 GHz. This is a specific band of some countries, such as Russia and neighboring countries (the countries of the authors), so the analysis is quite specific to that region. The authors claim that WRC'23 will define the loss of that band, however in most of countries that band cannot be for 5G due to national and regional regulation.
Since the paper is quite specific to the example of that band, I think that the authors should extend the results to other parameters, for example to analyse the problem with variable height of antennas (in Fig. 8 and 9, for example), so that the height of the antenna is one of the parameter of the analysis and the results are function of the height. Otherwise, the paper risks to be exclusively for the local case of Russia (with antennas of 20 and 60 meters).
Sometimes the results are quite obvious and the conclusions as well.
Please, consider that there are already commercial solutions where backhaul and fronthal is shared (some micro-cells). Please, refer those solutions in your paper.
Please, refer also other research papers where this problem is discussed, such as:
Yang, Di Yin, Xia Song, Xiaoming Dong, Gunasekaran Manogaran, George Mastorakis, Constandinos X. Mavromoustakis and Jordi Mongay Batalla, Security situation assessment for massive MIMO systems for 5G communications. Elsevier Future Generation Computer Systems, Sept. 2019. DOI: 10.1016/j.future.2019.03.036.
Author Response
Hello! Thank you for your review, please find the replies to your comments below:
Since the paper is quite specific to the example of that band, I think that the authors should extend the results to other parameters, for example to analyse the problem with variable height of antennas (in Fig. 8 and 9, for example), so that the height of the antenna is one of the parameter of the analysis and the results are function of the height. Otherwise, the paper risks to be exclusively for the local case of Russia (with antennas of 20 and 60 meters).
This is study doesn't consider the Russian case at all, all parameters including antenna heights (20 and 60 meters) were provided by the International Telecommunication Union in the frames of the WRC-23 study cycle and all other countries use exactly the same antenna heights in their study. Please refer to the ITU-R document with FS characteristics that I've attached provided by Working Party 5C of ITU-R. So we are not using any Russian-centric case here.
Please, consider that there are already commercial solutions where backhaul and fronthal is shared (some micro-cells). Please, refer those solutions in your paper.
There are no commercial solutions of 5G in the 6425-7125 MHz band, this band is not yet identified for IMT and even 3GPP hasn't finished with characteristics to this band. So far there is only one prototype from Huawei of 5G equipment to this band which is for a macro deployment.
The paper presents an analysis for haring fronthaul i backhaul in 5G in the band of 6.4-7.1 GHz. This is a specific band of some countries, such as Russia and neighboring countries (the countries of the authors), so the analysis is quite specific to that region. The authors claim that WRC'23 will define the loss of that band, however in most of countries that band cannot be for 5G due to national and regional regulation.
WRC-23 will not define the loss of that band for Fixed Service, at the same time it will globally identify this band for IMT-2020 i.e. 5G. Some countries will join that decision, some will not, that we will know only in WRC-2023. It is true that not all countries have plans for 5G in that band, for example, CITEL countries (USA, Canada, and Latin American countries) will use this band for Wi-Fi instead. But that doesn't change the fact that WRC-23 will identify that band for 5G and many countries in the world will join that decision, so far we may clearly see that China, Russia, and also many countries from European Union and Africa are planning to use that band for 5G.
Please, refer also other research papers where this problem is discussed, such as:
Yang, Di Yin, Xia Song, Xiaoming Dong, Gunasekaran Manogaran, George Mastorakis, Constandinos X. Mavromoustakis and Jordi Mongay Batalla, Security situation assessment for massive MIMO systems for 5G communications. Elsevier Future Generation Computer Systems, Sept. 2019. DOI: 10.1016/j.future.2019.03.036.
As far as I can see from the description of that paper, it doesn't provide any compatibility study of 5G with other services, so I don't see how it can be referenced.

Reviewer 3 Report
This paper presents sharing studies between 5G networks and point-to-point fixed service in the 6425-7125 MHz band. Besides this work is an extension of a previous work, you must add SoA section. You should comment each figure and extend the explanation of the results.
Author Response
Thank you for the review, we have extended the explanation of the results and added some additional information to the article. All the differences compared to the previous article were explained. We inserted it after the Introduction section and highlighted it with yellow color.
Best regards,
Alexander
Round 2
Reviewer 1 Report
Please, for the equation (1) check if it is correct, explain values used in this equation and give the reference. The calculated value seems to be too high.
There is still inconsistency between values given in the Table 5 and Figure 9. Please, correct values in the Table 5 or use proper chart.
Please, rewrite exponents in the equation (7), e.g., (ACS/10) - if it is an exponent of 10. And the same with (ACLR/10).
Please, correct grammar and style errors:
- rewrite the sentence in line 307 (which means that and for the ...),
- in line 310 the sentence “kept in mind that that” must be corrected as “kept in mind that”.
Author Response
Thank you for your comments.
Please, for the equation (1) check if it is correct, explain values used in this equation and give the reference. The calculated value seems to be too high.
The value is correct 41 dBm/33 MHz, that's just 12.5 Watts, not high for a base station, I have added the explanation under the expression how it was obtained.
There is still inconsistency between values given in the Table 5 and Figure 9. Please, correct values in the Table 5 or use proper chart.
Updated Table 5 and Figure 9, now the should look fine.
Please, rewrite exponents in the equation (7), e.g., (ACS/10) - if it is an exponent of 10. And the same with (ACLR/10)
It's a 10^(ACR/10), I have rewrote the formula to be it more clear.
Please, correct grammar and style errors:
Done the correction, thank you!
Reviewer 2 Report
The authors have not provided satisfying changes.
I can agree that 20 and 60 meters selection is ok, on the light of the document presented by the authors. Thank you.
For me the most important point which should be considered in the paper is the comparison with other systems that integrate fronthaul i backhaul. Obviously, these other systems work in other frequencies (6.4-7.1 GHz is a marginal example and it is even not clear if will be implemented for 5G). However, I think that the fact that there exist systems that integrate fronthaul i backhaul limits very much the innovation of this paper. Therefore, I think that it should be included in the paper, pointing out the differences and showing why the proposal of the authors is better.
Author Response
I understand your idea, I have added the following information to the introduction section to clarify that moment, in addition I have uploaded 5G characteristics for 6425-7125 MHz band provided by ITU-R:
At present there are no commercial solutions of 5G base stations or user equipment and characteristics of them are preliminary provided by 3GPP and ITU-R (see 3GPP TS.38.104 v. 16.6.0 (2020-12) «NR; Base Station (BS) radio transmission and reception», 3GPP TS.38.101-1 v.16.6.0 (2020-12) « NR; User Equipment (UE) radio transmission and reception; Part 1: Range 1 Standalone» and Document 5D/TEMP/422). The architecture options of 5G base stations that are defined in specifications 3GPP TS 38.104 and 3GPP TS 38.401 as well as in ITU-T Rec. G.8300 (05/2020) allow using the integration of fronthaul and backhaul in one base station for the bandwidth 6425-7125 MHz (FR1). The architecture solutions that integrate both fronthaul and backhaul in one BS do not influence the issue of spectrum sharing and electromagnetic compatibility, since the architecture is related to network design rather than compatibility issues. Therefore, the obtained by the authors results are applicable for any architecture solution of 5G base stations.
